# Enterococcal Urinary Tract Infections: A Review of the Pathogenicity, Epidemiology, and Treatment

**DOI:** 10.3390/antibiotics12040778

**Published:** 2023-04-19

**Authors:** Alia Codelia-Anjum, Lori B. Lerner, Dean Elterman, Kevin C. Zorn, Naeem Bhojani, Bilal Chughtai

**Affiliations:** 1Department of Urology, Weill Cornell Medical College, New York Presbyterian Hospital, New York, NY 10065, USA; alc4017@med.cornell.edu; 2Department of Urology, VA Boston Healthcare System, Boston, MA 02132, USA; 3Division of Urology, Department of Surgery, University Health Network, University of Toronto, Toronto, ON M5T 2SB, Canada; 4Division of Urology, Centre Hospitalier de l’Université de Monstréal, Montreal, QC H2X 0A9, Canada

**Keywords:** enterococcus, urinary tract infection, resistance, vancomycin resistant enterococcus, biofilm

## Abstract

Urinary tract infections (UTIs) are among the most common causes of infections worldwide and can be caused by numerous uropathogens. *Enterococci* are Gram-positive, facultative anaerobic commensal organisms of the gastrointestinal tract that are known uropathogens. *Enterococcus* spp. has become a leading cause of healthcare associated infections, ranging from endocarditis to UTIs. In recent years, there has been an increase in multidrug resistance due to antibiotic misuse, especially in enterococci. Additionally, infections due to enterococci pose a unique challenge due to their ability to survive in extreme environments, intrinsic antimicrobial resistance, and genomic malleability. Overall, this review aims to highlight the pathogenicity, epidemiology, and treatment recommendations (according to the most recent guidelines) of enterococci.

## 1. Introduction

Urinary tract infections (UTIs) are among the most common causes of infections across all genders and age groups worldwide [1]. Over 404.6 million people across the planet were diagnosed with UTIs in 2019, which accounted for over 200,000 deaths [2]. The financial burden of UTI-associated hospitalizations is substantial, with upwards of 2.8 billion dollars spent in the United States in 2011 [3]. Data collected from the Global Health Data Exchange from 1990 to 2019 revealed that the rate of infection, mortality, and disability-adjusted life-years has increased worldwide [2]. Given the strain UTIs and their associated sequelae put on the health of individuals, hospital systems, and populations, continued efforts to understand and mitigate the occurrence remain vital. UTIs can be further classified as either uncomplicated or complicated. An uncomplicated or simple UTI is defined as an infection in the lower urinary tract system in either a male or non-pregnant female patient [4]. Complicated UTIs are associated with atypical organisms, patients considered high risk (pregnancy, comorbidities, immunosuppression, etc.), or involve the upper urinary tract. The most common bacterial pathogen responsible for UTIs is *Escherichia (E.) coli*, making up nearly 80% of infections. The other 20% are comprised mostly of *Klebsiella *(*K.*)* pneumonia*, *Proteus *(*P.*)* mirabilis*, *Enterococcus *(*E.*)* faecalis*, and *Staphylococcus *(*S.*)* saprophyticus* [4].

The focus of this paper is on enterococci, which are Gram-positive, facultative anaerobic commensal organisms of the gastrointestinal tract, and known uropathogens. *Enterococcus* spp. has become a leading cause of healthcare-associated infections ranging from endocarditis to UTIs [4]. Infections due to enterococci pose a unique challenge due to their ability to grow in extreme environments, as well as intrinsic and multidrug antibiotic resistance, making them a topic of interest [5]. In this review, we present an overview of enterococcal urinary tract infections and treatment considerations.

## 2. Pathogenicity

Of the known enterococcus species, the majority of urinary tract infections are due to *E. faecalis* and *E. faecium*, which have multiple mechanisms that increase their pathogenicity [4,5]. These mechanisms include biofilm formation and virulence factors.

### 2.1. Biofilm

Biofilm formation by enterococcus has been observed and studied over the last 40 years. First described in the mid 1980s, there has since been robust investigation into how they develop and impact virulence and resistance patterns [6]. We know that biofilm formation is a multifactorial process that enables evasion of the host defenses and enhances bacterial virulence and antibiotic tolerance [6,7,8,9]. Ch’ng et al. described four stages of biofilm development by *E. faecalis*: attachment, microcolony formation, biofilm maturation, and dispersal [7]. Attachment is facilitated by surface adhesins (aggregation substance and enterococcal surface protein), proteases, and glycolipids. Once securely attached, *E. faecalis* bacteria multiply and secrete a biofilm matrix that forms microcolony aggregates. This biofilm matrix then matures with the production of extracellular matrix components, such as extracellular DNA, polysaccharides, lipoteichoic acid, extracellular proteases, modified lipids, and glycoproteins. As the microcolony grows and matures, the colony endures local nutrient deficits, crowding, hypoxia, and waste accumulation, which creates environmental stress [7,8]. The resultant stress response prompts a gene expression shift from maturation to dispersal, which is generally regarded as the final step of biofilm development. In dispersal, the core of the biofilm liquefies and the microcolony wall is disrupted, allowing individual bacteria to escape and form new colonies [7,8]. While the process of biofilm formation of *E. faecium* closely resembles that of *E. faecalis,* it is not identical. Although both *E. faecalis* and *E. faecium* produce biofilm, their mechanism of protection is different. *E. faecalis* produces a thick film that is difficult to penetrate, while the film *E. faecium* produces contains antibiotic-resistant genes [7]. Adherence and biofilm formation appear to be a hallmark of *Enterococcus* spp., especially in urinary isolates [6].

Biofilms pose further difficulty in treatment as they invite polymicrobial colonization with other species, namely *E. coli*, which has been found in co-isolates with *E. faecalis* in urinary tract infections [7]. In this communalistic relationship, *E. Faecalis* increases the virulence of *E. coli* through immunomodulation and suppression. In a study by Tien et al., it was observed that *E. faecalis* is able to subvert the recruitment and activation of immune cells such as macrophages, by preventing nuclear factor kappa B (NF-κB) signaling [9]. Another method of immunomodulation is through the secretion of gelatinase, which works by cleaving complement components (C3, C3a, and C5a). In doing so, *E. faecalis* is able to evade the innate immune system. Furthermore, Tien et al. investigated this communalistic relationship and its role in catheter-associated urinary tract infections (CAUTI). They found that multimicrobial CAUTI that contained both *E. faecalis* and *E. coli* showed a lower presence of macrophages than in CAUTI caused by only *E. coli*.

### 2.2. Virulence Factors

Enterococcus also has a variety of other methods that increase its pathogenicity, known as virulence factors [6,7]. Virulence factors are molecules that increase pathogenicity and assist in the survival and colonization of bacteria in the host environment [10]. Some known virulence factors found in urinary isolates of *Enterococcus* spp. include aggregation substances, enterococcal surface proteins, pilin gene clusters (PGCs), collagen binding protein, TcpF, and gelatinase [6,9,11,12,13,14,15,16,17,18,19,20]. Enterococcal surface proteins (*Esp*) are known facilitators of biofilm formation and have been shown to promote the primary adhesion. *Esp* has been found in both *E. faecalis* and *E. faecium* [6,11,16]. A study by Shankar et al. further examined the role of *Esp* in *E. faecalis*-mediated UTIs and found that *Esp* has a vital role in colonization [11]. Aggregation substance (AS) is a necessary surface adhesion that mediates adhesion to host cells, is responsible for bacterial aggregation, and promotes cell conjugation with pheromone-responsive plasmids [6,9,12,13]. Interestingly, some species of enterococci express a larger volume of one type of virulence factor compared with others. For example, in urinary isolates of *E. faecalis*, a high frequency collagen-binding protein was observed [12]. Table 1 lists common virulence factors and their role in increasing enterococcal pathogenicity.

## 3. Resistance Patterns

*Enterococcus* spp. are resistant and tolerant to a broad spectrum of antibiotics, which poses a treatment challenge. They have also been known to acquire and share resistance to antimicrobials with ease [22]. This characteristic of *Enterococci* is at least partly due to the fact that they have extremely malleable genes, which allows them to readily acquire mobile genetic elements, create hybrid genomes with other enterococci, and transfer genes across species [23]. Additionally, the lack of genomic defense mechanisms, such as resistance-modification systems and CRISPR-Cas, appears to increase the acquisition of antibiotic resistance genes.

Horizontal gene transfer is known to facilitate the movement of genetic information though plasmids and transposons [23]. Mainly in *E. faecalis*, although seen in *E. faecium*, pheromones are produced, which stimulate pheromone-responsive plasmids to begin the conjugative process [24]. Additionally, pheromone responsive plasmids play an important role in providing *E. faecalis* with numerous accessory genes. Non-pheromone-dependent plasmids are also responsible for sharing genetic information through conjugation. These plasmids have been associated with the dissemination of antibiotic resistance genes to other bacteria outside of the enterococcal genus. Furthermore, it has been shown that a plasmid is capable of carrying multiple antibiotic resistance genes [23]. These genes have been linked to multiple virulence factors and antibiotic resistance. It is believed that these types of plasmids evolved to have host specificity in enterococcal species, as certain plasmids were only found in one *E. faecalis* and not *E. faecium*, and vice versa. Transposons have also been linked to the resistance of multiple drugs including tetracycline, gentamicin, and glycopeptides, by encoding for antibiotic-resistant genes [24,25].

The resistance of *Enterococcus* spp. to vancomycin because of gene clusters such as *van*A, B, C, D, and E is particularly notable, and can develop over a short period of time [26]. These vancomycin-resistant gene clusters facilitate resistance by altering peptides that form the cell wall precursors to which Vancomycin binds. The altered peptide termini disrupt vancomycin from properly binding, resulting in vancomycin having a weak affinity [26]. Interestingly, Swaminathan et al. showed that the prevalence of specific phenotypes of *E. faecalis* and *E. faecium* vary in different parts of the world, with *van*A being isolated in Europe and North America, and *vanB* found in Australia and Asia [26]. The *van*A phenotype has been shown to exhibit resistance against both vancomycin and teicoplanin. *E. gallinarum* and *E. casseliflavus/flavescens* make up a small portion of enterococcal infections and have an intrinsic, low-level resistance to vancomycin due to the expression the *van*C gen [27].

Vancomycin-resistant enterococci (VRE) infections are often nosocomial, with *E. Faecium* being the most common isolate, followed by *E. faecalis* [28]. From 2000 to 2006, the amount of VRE infections increased from about 10,000 to 21,000 [25]. Unfortunately, enterococci have shown the ability to share antibiotic-resistant genes through horizontal gene transfer [23]. The ability of enterococcus to transfer *van*A to *S. aureus* was observed both in vitro and in vivo. Additionally, there have been cases of methicillin-resistant Staphylococcus aureus gaining vancomycin resistance from enterococci [29,30]. The implication of this trait makes enterococcus even more worthy of investigation.

*Enterococci* also show a natural resistance to other antibiotics, including oxazolidinones, quinolones, and most β -lactams, such as cephalosporins. Table 2 provides a brief overview of the various compounds to which enterococci have shown resistance, and how.

## 4. Epidemiology

### 4.1. Ambulatory Population

While UTIs caused by enterococcus largely remain nosocomial, they are also responsible for a small percentage of community-associated infections [34]. In a multicenter cross-sectional study by Malmartel et al., urine cultures were collected from 1119 patients seen in outpatient offices and were then analyzed [35]. Of those cultures, 7% identified *Enterococcus* spp. Seitz et al. prospectively collected urine cultures from 423 female patients exhibiting symptoms of acute uncomplicated cystitis, with *E. faecalis* isolated in 10.2% of samples [36]. While not as high an incidence, Laupland et al. found that 5.3% of their ambulatory patients with UTIs had *Enterococcus* spp. isolated from their urine [37].

Interestingly, Silva et al. found the prevalence of UTI due to *E. faecalis* to be higher in men (8.8%) compared with women (1.8%) [38]. This study suggests that a patient’s sex is an important consideration when treating UTIs. In a retrospective, multicenter study by Salm et al., 102,736 urine cultures were collected and analyzed from males in an outpatient setting [39]. *E. faecalis* isolates were identified from 16.5% of cultures, of which 22.9% were considered to be polymicrobial infections. The frequency of non-*E. Coli* bacteria responsible for UTIs in males is noteworthy and can affect the choice of antimicrobial therapy. While it has been shown that bacterial resistance is higher among older patient populations, this rate increases even more in male patients, further driving this point home. Why men are more susceptible to *Enterococcus* infections than women is not known, but several hypotheses exist. First, men have prostates that may harbor bacteria and form small micro-abscesses. The translocation of bacteria from the intestinal tract can seed the prostate tissue [40]. Indeed, microbiota that exist in the prostate have been suggested to impact prostate cancer, and enterococcus has even been found in sperm [41]. Second, prostate stones are not an uncommon finding [42]. It is possible that these stones become secondarily infected, and a biofilm develops both on the stone and in the cavity where the stone exists. While not proven, research among urologists suggests that when a prostate stone nidus is identified, removing the stones and opening the crypts could reduce recurrent infections.

Research has shown that *E. faecium* strains isolated from community-associated infections are different at their core genome from those isolated from nosocomial infections [5]. They can be divided into two clades, with Clade A being associated with healthcare-acquired infections and Clade B being associated with community infections. VRE has been almost exclusively associated with hospital infections, but one cross-sectional study of 100 patients sought to identify the presence of VRE among ambulatory patients [43]. Colonization was seen in three patients, with one patient having no recent antibiotic treatment or exposure to a healthcare facility. The implications of VRE extending beyond health centers pose a substantial risk to public health by limiting treatment options.

### 4.2. Hospitalized Population

According to the International Society for Infectious Diseases and the Center for Disease Control, UTIs are the fifth most common infection in hospitalized patients and account for 12.9% of nosocomial infections [44,45]. Hospital-acquired UTIs are associated with catheterization among patients, length of hospitalization, and the multi morbid status of patients. Enterococcal prevalence in healthcare settings can be attributed to the bacterial ability to survive on multiple surfaces for long periods of time and in harsh, heavily disinfected environments [46]. *Enterococcus* spp. contributes to over 30% of nosocomial UTIs and has been identified as the second leading pathogen in CAUTI [5,47]. Catheters are an ideal setting for bacterial growth as they provide a surface for biofilm adhesion and disrupt the bladder environment. A study by Guiton et al. investigated the inflammatory impact that catheters have on the urinary tract and their role in promoting infection [48]. This study suggested that the introduction of a catheter and subsequent irritation causing inflammation was what helped enterococcus establish an infection. They found that inflammation secondary to *E. faecalis* infection in a non-instrumented bladder was often minimal and was easily cleared by the bladder; however, when a catheter was added, the outcomes were very different. The degree of inflammation generated by the catheter itself, combined with the presence of *E. faecalis* bacteria, induced a much more significant infection. Additionally, the authors showed that the host immune response on its own was not enough to clear the infection, which further suggests that patients undergoing immunosuppressive therapies may have an increased risk of colonization and infection [48].

Adding comorbidities to hospitalized patients further increases the risk of symptomatic enterococcus infections. Shin et al. analyzed urine cultures from 301 hospitalized patients with UTIs who were also suffering from neurological diseases [49]. Of these patients, 272 (90.4%) of the infections were considered hospital-acquired, and 142 (47.3%) were associated with catheters, while 159 (52.8%) were not. Overall, *Enterococcus* spp. accounted for 27.2% of the total infections and was the primary uropathogen in 47.2% (142) of CAUTI cases. Finally, enterococcal UTIs have also been shown to be a source of endocarditis and can lead to bacteremia [50]. These studies highlight how certain patient populations in hospital settings are at particular risk for significant enterococcal infections.

### 4.3. At Risk Population (Immunocompromised and Comorbidities)

As mentioned in the prior section, patients with comorbidities are at higher risk. When those patients are also immunocompromised, for example with organ transplant and cancer, the development of complicated UTIs is even more significant [26,51]. In renal transplant patients, UTIs are one of the most common afflictions patients encounter in their first-year post-transplant, with nearly half of patients developing bacteriuria [51]. The major causative organisms, *E. coli* and enterococcus (35% of the total UTIs), also demonstrated high rates of multidrug resistance at 36%. Given the risk to these vulnerable patients, the authors advise treating ASB in renal transplant patients, a practice nearly always discouraged in the general population. Swaminathan et al. further supported the practice of treating transplant patients with ASB [26]. In the case of enterococcus, colonization frequently leads to symptomatic infection and when it does, the risk of VRE is higher. As has been discussed, the limited treatment options for VRE and the high rate of mutagenicity with enterococcus supports the approach of attempting prevention and early treatment, even in the absence of symptoms.

### 4.4. Asymptomatic Bacteriuria (ASB)

Asymptomatic bacteriuria in the general population is quite common. Often patients in hospitals who, for whatever reason, have urine checked will have colonized enterococcus in their urinary tract. For years, many organizations have stood by the message of not treating asymptomatic patients [44]. That being said, what constitutes symptoms can vary between organizations, as well as clinical providers, and even patients themselves. It has long been known that the overuse of antibiotics to treat colonization that is not causing symptoms contributes to the creation of antibiotic-resistant bacteria. Given the limited number of antibiotics available to treat resistant organisms and the lack of support to develop new antibiotics, the need to control resistance is vitally important in population health [52]. A retrospective multicenter study by Lin et al. aimed to explore if UTIs and ASBs were being managed appropriately [47]. Of the 339 patients, 183 were classified as having ASB and 156 had symptomatic UTIs. Overall, 83 patients were not managed according to their diagnosis. Of the patients with ASB, 60 (32.8%) were inappropriately given antibiotics and of the patients with UTIs, 23 (14.7%) were not given antibiotics. This misuse of antibiotics drives the increase in multidrug resistance, a growing crisis that all clinical providers need to help avoid.

## 5. Treatment

When managing urinary infections caused by enterococcus, identifying susceptibility is extremely important when choosing the appropriate antibiotic therapy, due to enterococci’s propensity for multidrug resistance. Knowing which species is the cause of infection should also be considered when formulating a treatment plan, as resistance patterns and virulence factors vary. Antibiotics should not be prescribed without clear clinical evidence that the patient has a symptomatic UTI and not ASB. As highlighted earlier in the review, only certain populations should be given antibiotics for ASB [26,51].

Despite some resistance, ampicillin has been shown to be effective due to the high concentration of the antibiotic in urine. One study found no difference in treating ampicillin-resistant *E. faecium* with either amoxicillin or nitrofurantoin [53]. Additionally, UTIs due to VRE showed susceptibility to nitrofurantoin in vitro [38,53,54]. For the treatment of uncomplicated infections, the use of amoxicillin, fosfomycin, or nitrofurantoin is preferred [55]. If susceptibility indicates otherwise, alternative treatment includes either ampicillin, fluoroquinolones, oxazolidinones, vancomycin, or daptomycin. For complicated infections, intravenous ampicillin, fluoroquinolones, oxazolidinones, vancomycin, or daptomycin can be used [55]. However, in severe infections, ampicillin is recommended to be used in conjunction with either streptomycin or gentamicin [31]. This is further summarized in Table 3.

Natural remedies have long been lauded for treating or preventing UTIs [56,57]. With the rise in multidrug-resistant infections, consideration should be given to alternative treatment options. In an AUA guideline regarding the management of recurrent UTIs in females, studies of non-antibiotic prophylactic treatments were identified and reviewed [56]. Cranberries have been believed to prevent UTIs as they contain proanthocyanins (PACs) [57,58]. It is thought that this compound prevents the adhesion of bacteria to the lining of the bladder. Multiple studies have assessed the efficacy of cranberry in various forms such as juices, powders, and tablets. A Cochrane database systematic review of randomized control trials in the use of cranberry as a prophylactic treatment of UTIs was performed [56]. This review identified three studies that noted no significant difference in the prevention of UTIs when comparing cranberry and antibiotics. One disadvantage is that the formulation of these cranberry products is not uniform in their dosing. Further trials with uniform dosing must be performed to prove the validity of cranberry as a treatment as the ideal dosage for cranberry has not yet been identified.

In a study by Wojnicz et al., multiple tests were conducted to identify the effect of cranberry extract on urinary isolates of *E. faecalis* [58]. To test if cranberry affected biofilm formation, 10 strains of *E. faecalis* were grown on agar plates with cranberry extract and without it as a control. The presence of biofilm was then checked at three different time points: 24 h, 48 h, and 72 h. At the 24 h mark, 5 out of the 10 strains showed decreased biofilm production, and at the 48 h mark, 7 out of the 10 strains showed a decrease as well. By the 72 h mark, all strains showed a significant decrease in biofilm production when compared with the control plates. Additionally, the effect of cranberry extract was tested on the synthesis of known virulence factors. The synthesis of gelatinase was reduced in nearly 50% of strains and 40% of strains showed a significant reduction in both lecithinase and lipase. Although the role of cranberry in preventing enterococcal infections is limited, this study showed promising results [58].

Alternative remedies should be considered as an option for patients who are at a higher risk of developing severe infections, such as pregnant patients, renal transplant patients, or for patients who may be catheterized for long periods of time to prevent CAUTI.

### Catheter Associated Urinary Tract Infection (CAUTI)

Prevention is the most effective way of stopping CAUTI from occurring. This includes avoiding unnecessary catheterization and immediate removal of catheters when no longer clinically needed [59]. One study found VR *E. faecalis* to be reduced by 97% after coating silicon catheters with non-leachable cationic film coatings, which may be another method of prevention [7]. Another method of preventing the formation of biofilm on catheters is the use of low-energy surface acoustic waves (SAW) [60]. This mechanism works through the use of a piezoelectric actuator that is attached to the catheter. The actuator sends vibrations along the catheters surface at a frequency between 100 to 300 kHz, thus disrupting adhesion. In a study by Hazan et al., catheters were incubated over a period of 3 days with a medium that contained multiple organisms, including *E. faecalis* and *E. coli*. At the end of the 3 days, half of the catheters were treated to SAW. The catheters that underwent SAW showed a significant reduction in biofilm coating when compared with the control catheters. In further testing, SAW was shown to interfere with adhesion in *E. coli* [60]. If a patient has a CAUTI or colonization, catheter removal, when possible, has been shown to resolve the infections [55]. However, in patients who have CAUTI but still require catheterization, routine replacement of the catheters is strongly recommended [59].

## 6. Conclusions

Over the years, UTIs due to enterococci have become more prevalent in both nosocomial- and community-acquired infections. Enterococcus is unique in its ever-increasing pathogenicity through a large variety of mechanisms, especially biofilm formation and genetic malleability. Despite the fairly well understood basics of biofilm development among several enterococcal species, there is still much that is unknown. The implications of their pathogenicity and the subsequent ripple effect of these bacteria can cause pose a significant risk to population health. As antibiotic resistance increases, concurrently with the decreasing development of new therapeutics, it is essential that focus and investigation into these unique processes continue. When managing enterococcal infections, increased attention and consideration must be given when formulating a treatment plan.

## Figures and Tables

**Table 1 antibiotics-12-00778-t001:** Known virulence factors of enterococcus.

Virulence Factor	Function
Aggregation substance [6,9,12,13]	- Responsible for bacterial aggregation and helps facilitate conjugation - Mediates adhesion to host cells and extracellular matrix
Collagen binding protein [12]	- Facilitates adhesion to extracellular matrix and type 1 collagen
Cytolysin [14,15]	- Toxin that is able to lyse various cell types through pore formation - Encoded through plasmids
Enterococcal surface protein [7,16]	- Promotes initial adherence to surfaces and biofilm formation - Associated with bacterial colonization and persistence
Gelatinase [9,17]	- Able to break down various substrates including complement proteins and collagen - Assists with biofilm formation
Hyaluronidase [18]	- Breaks down hyaluronic acid to increase permeability of connective tissues
Pilin gene clusters (PGCs) [6,19]	- Encodes gene for pili formation, which assists with biofilm formation and adhesion to host cells - *E. faecalis* and *E. faecium* express different PCGs, however, they perform similar roles in adherence and biofilm formation
TcpF [9,20,21]	- Suppresses Toll-like receptors (TLR) from producing cytokines by interfering with the signaling pathway

**Table 2 antibiotics-12-00778-t002:** Antibiotic mechanisms and enterococcal resistance.

Antibiotic	Mechanism of Action	Mechanism of Resistance
Aminoglycosides [25,31,32]	Interferes with bacterial protein synthesis by binding to the 30S ribosome subunit	Aminoglycoside modifying enzymes, the only exceptions being streptomycin and gentamicin Streptomycin resistance occurs in 2 ways: enzymatic inactivation or ‘absolute’ inhibition Gentamicin resistance occurs due to the enzyme AAC(6′)-Ie/APH(2′), which contains 6′ acetyltransferase and 2′ phosphotransferase
β-lactam [5,25,26,31]	Ampicillin and penicillin function against enterococci infections by inhibiting the synthesis of peptidoglycan, which is an essential part of the cell wall.	*Enterococci* can resist the effects of these β-lactam compounds through the expression of the *pbp5* gene This gene encodes for a penicillin-binding protein in which ampicillin and cephalosporins have a poor binding affinity. A two-component regulatory system consisting of IreK, a serine/threonine kinase, and IreP, a phosphatase, was shown to contribute to cephalosporin resistance
Daptomycin [5,25,31,33]	Disrupts bacterial cell membrane function causing loss of membrane potential	There are 3 genes associated with Daptomycin resistance: LiaF (part of the LiaFSR regulatory system), gpdD, and Cls (both involved in phospholipid metabolism Mechanism of resistance differs in both *E. faecalis* and *E. faecium,* but LiaFSR seems to play a role in both
Glycopeptides [5,23,24,25,31]	Binds to precursors and inhibits the synthesis and permeability of both the bacterial cell wall and membrane	Encoded in van clusters that alter cell wall precursors that vancomycin binds
Oxazolidinones [25,31]	Binds to the 23SrRNA and disrupts protein synthesis	Resistance commonly occurs due to mutations in the gene that encodes for 23SrRNA, preventing antibacterial binding
Quinolones [5,31]	Targets the enzymes necessary for transcription and replication	Quinolone resistance commonly occurs by mutations in the target genes that decrease the binding affinity
Rifampicin [31]	Inhibits mRNA transcription by binding to the β-subunit of RNA polymerase, encoded by *rpoB* gene	Resistance of this drug occurs through mutations in the *rpoB* Gene
Trimethoprim and sulphamethoxazole [31]	Inhibits the enzymes required in the folate synthesis pathway, as many bacteria are unable to exogenously acquire folate	*Enterococci* can exogenously acquire folate

**Table 3 antibiotics-12-00778-t003:** Therapeutic agents for the treatment of simple and complicated cystitis.

**Simple Cystitis**	
Preferred oral agents	Amoxicillin, nitrofurantoin, and fosfomycin
Alternative oral agents	Levofloxacin and linezolid
Intravenous alternative agents	Ampicillin, vacomycin, daptomycin, or linezolid
**Complicated Cystitis**	
Preferred intravenous agents	Ampicillin *
Alternative intravenous agents	Fluoroquinolones, oxazolidinones, vancomycin, or daptomycin

* Ampicillin is recommended to be used in conjunction with either streptomycin or gentamicin. Adapted from references: [31,55].

## Data Availability

Not applicable.

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
