# Peer review of "Enterococcal Urinary Tract Infections: A Review of the Pathogenicity, Epidemiology, and Treatment"

_antibiotics, 2023, doi:10.3390/antibiotics12040778_

Round 1

Reviewer 1 Report

The current article (antibiotics-2299450) describes the epidemiology, pathogenicity and treatment option of Enterococcal UTI. The theme and information of this review will be of great interest for readers and scientific community to know the latest pattern of drug resistance, virulence and treatment of Enterococci associated with UTI.

Comments:

  • There are numerous articles on this topic, What kind of new information will be added by the current review article?
  • The author has given the information that Enterococcus spp. are the cause of approximately 110,000 UTIs in the 16 United States every year and are commonly associated with nosocomial infections. Why the author has given reference of United States? Generalized statement on the global cases should be added.
  • The global epidemiology of enterococcal infections should added in tabulated form supported by proper reference.
  • The virulence factors has been mentioned in table 2 but no proper description is given in the main text. This section need thorough revision.
  • I would suggest to include antibiotic resistance as separate section, not under the pathogenicity subsection.
  • Time line (year) graph of development of antibiotic resistance in Enterococci may be included.
  • The scientific names should be in italic form throughout the manuscript.
  • Latest references may be included in the introduction section.
  • Urinary tract infections once defined as UTI in the text, should be written as UTI later on.
  • Table 1,2: Additional column may be added for citing references
  • The title of all tables should be rephrased scientifically
  • The available treatment options for Enterococcal infections should also be given in tabulated form.

Author Response

Reviewer 1:

  1. There are numerous articles on this topic, What kind of new information will be added by the current review article? – We agree that this review does not necessarily describe something new that cannot be found elsewhere. However, the topic was requested and as we wished to support the journal, we submitted an article, as requested.
  2. The author has given the information that Enterococcus spp. are the cause of approximately 110,000 UTIs in the 16 United States every year and are commonly associated with nosocomial infections. Why the author has given reference of United States? Generalized statement on the global cases should be added. - done
  3. The virulence factors has been mentioned in table 2 but no proper description is given in the main text. This section need thorough revision. - done
  4. I would suggest to include antibiotic resistance as separate section, not under the pathogenicity subsection. – done
  5. Time line (year) graph of development of antibiotic resistance in Enterococci may be included. – this was not the focus of our paper. We are urologists and focused on current management. If a full description of individual antibiotics and their resistance history is desired, we will require more time to create this.
  6. The scientific names should be in italic form throughout the manuscript. - done
  7. Latest references may be included in the introduction section. – done
  8. Urinary tract infections once defined as UTI in the text, should be written as UTI later on. – done
  9. Table 1,2: Additional column may be added for citing references – this is not a systematic review and we are not comparing studies against each other. What is listed is not unique and not based on individual trials/studies. We prefer to have the focus be on the content.
  10. The title of all tables should be rephrased scientifically – we are unsure what the reviewer is asking.
  11. The available treatment options for Enterococcal infections should also be given in tabulated form. – done

Reviewer 2 Report

antibiotics-2299450-peer-review-v1

 Title of the manuscript is somewhat redundant with title of the special issue; to my opinion, it should be further modified.

The abstract is quite general; it should cover some specific information compiled within manuscript. Particularly, the highlights on pathogenicity, epidemiology, and treatment strategies for the enterococci discussed in manuscript.

Add at least one figure illustrating the focus of this manuscript e.g. a schematic epidemiological pathway of enterococci.

Table legends: (1) Use plural for mechanism. (2) Extend the title further elaborating the information presented in table.

Extra spaces from the tables should be removed to comprehensively shorten their sizes.

L40-42 and throughout the manuscript: Italicize the technical names of microorganisms, wherever used in ms.

L57: Ch’ng; please correct accordingly.

L71-73: use appropriate references.

Paragraph with L56-73 is exactly repeated in L74-91.

Section 2.1, Biofilm formation: is this whole process reported by 1-2 study only? If so, then better to wait for validation from the other researchers and if often reported, then use those multiple references should be used here.

L75, L120 and L164: et al.  et al  and et al.,   : be consistent with format throughout the ms, probably et al.

L125: expression of

L165: correct the sentence linguistically.

L200-202: add reference accordingly.

L202-203: Enterococci’s and Bacteria’s as enterococcal and bacterial

L212: full stop before However or however

L224: CAUTI, mention the abbreviations on their 1st use in ms.

L234-235: add reference accordingly.

L299-302: the previous reference continues or some other study? Please mention accordingly.

Author Response

Reviewer 2:

  1. Title of the manuscript is somewhat redundant with title of the special issue; to my opinion, it should be further modified. – we are not sure what the special issue title is. We were asked to submit a paper with this topic and used the title that was given to us.
  2. The abstract is quite general; it should cover some specific information compiled within manuscript. – as this is a review article, we are not sure how to enhance the abstract. The abstract is brief, but does state what the review is about.
  3. Add at least one figure illustrating the focus of this manuscript e.g. a schematic epidemiological pathway of enterococci.- this was not discussed in the paper and is beyond the ask, which was specifically about UTI’s.
  4. Table legends: (1) Use plural for mechanism. (2) Extend the title further elaborating the information presented in table. – we are unsure what the reviewer is asking. The tables are basic – virulence factors and treatment options. We are unclear how restating the table in the title will enhance understanding.
  5. Extra spaces from the tables should be removed to comprehensively shorten their sizes.- done
  6. L40-42 and throughout the manuscript: Italicize the technical names of microorganisms, wherever used in ms. – done
  7. L57: Ch’ng; please correct accordingly. – this spelling is correct
  8. L71-73: use appropriate references. – addressed
  9. Paragraph with L56-73 is exactly repeated in L74-91. – duplicate removed
  10. Section 2.1, Biofilm formation: is this whole process reported by 1-2 study only? If so, then better to wait for validation from the other researchers and if often reported, then use those multiple references should be used here. – discussion and references expanded
  11. L75, L120 and L164: et al. et al and et al., : be consistent with format throughout the ms, probably et al. – corrected
  12. L125: expression of L165: correct the sentence linguistically. – done
  13. L200-202: add reference accordingly.- done
  14. L202-203: Enterococci’s and Bacteria’s as enterococcal and bacterial – done
  15. L212: full stop before However or however - done
  16. L224: CAUTI, mention the abbreviations on their 1st use in ms. – done
  17. L234-235: add reference accordingly. – done
  18. L299-302: the previous reference continues or some other study? Please mention accordingly. – addressed

Reviewer 3 Report

The manuscript submitted gives a good overview on the role of Enterococci in urinray tract infections.

Two remarks:

The role of Enterococci in uncomplicated UTI in outpatients may be somewhat overemphasized. Some large surveys report a frequency of Enterococci of less than 5% (e.g. Naber et a.. Eur Urol 2008). 

For practical purposes, it should be explicitly mentioned that enterococci have a natural resistance against all cephalosporines. 

Author Response

Reviewer 3:

  1. The role of Enterococci in uncomplicated UTI in outpatients may be somewhat overemphasized. Some large surveys report a frequency of Enterococci of less than 5% (e.g. Naber et a.. Eur Urol 2008). – we have revised this portion and added the reference
  2. For practical purposes, it should be explicitly mentioned that enterococci have a natural resistance against all cephalosporines - done

A request from all the authors is a clarification on what exactly the journal was hoping to publish. We were asked to submit a publication with this title and topic, one that is a review in nature. There are certainly many publications and book chapters on this topic, making this article a summary of what is already available. It was our goal to provide a somewhat condensed, but informative, summary on enterococcus in urinary tract infections for the readers of this journal, but did not believe the intention was to expand to a full chapter-sized publication. We hope that this revised version will meet the desires of the Editors. If it continues to fall short, a clear explanation of what the journal would like would be appreciated, with an appropriate expansion of the deadline to allow the authors (all urologists) to elaborate on what we have submitted.

Thank you for the invitation and we look forward to your response.

Sincerely

Lori B. Lerner, MD, and team

Round 2

Reviewer 1 Report

I appreciate the authors for incorporation of suggested changes.

Reviewer 2 Report

Though, it looks strange that the authors do not know the title of the special issue they are going to publish in, the comments are largely responded.

The revised ms is improved.